# A Review on Recent Trends and Applications of IoT in Additive Manufacturing

**Bharat Kumar Chigilipalli** [1], **Teja Karri** [1], **Sathish Naidu Chetti** [1], **Girish Bhiogade** [1], **Ravi Kumar Kottala** [2] **and Muralimohan Cheepu** [3,4,*]

1. Department of Mechanical Engineering, Vignan's Institute of Information Technology (A), Visakhapatnam 530049, India
2. Department of Mechanical Engineering, National Institute of Technology Tiruchirappalli, Tiruchirappalli 620015, India
3. Department of Materials System Engineering, Pukyong National University, Busan 48547, Republic of Korea
4. STARWELDS Inc., Busan 46722, Republic of Korea
* Correspondence: muralicheepu@gmail.com

**Abstract:** The Internet of Things (IoT) is a new way of communicating that is changing the way things are monitored and controlled from a distance. Gradually, companies want to digitalize their production processes and implement control and monitoring systems on the shop floor. On the basis of the Industry 4.0 concept, internet features and database services have been incorporated into processes in order to reinvent manufacturing. This study proposes a proof-of-concept system for the management of additive manufacturing (AM) machines, where an internet integration of beacon technology in the manufacturing environment enables the rapid and intuitive interchange of production data retrieved from machines with mobile devices in various applications. Even though AM technologies can be used to customize the final product, they cannot be used to make a lot of 3D-printed jobs at once for commercial usage. Therefore, this research-based study aims to understand IoT technologies to improve the understanding and reliability of AM processes and 3D print smart materials in large quantities for manufacturers around the world. This study demonstrates the significance of the successful use of internet-based technologies in AM by examining its practical consequences in various fields. This paper gives an overview of IoT-based remote monitoring and control systems that could solve problems in AM, particularly in digital twin, human augmentation (HA), 3D bioprinters, 3D scanners, input parameters optimization, and electronics fields. IoT in AM makes production processes more efficient, reduces waste, and meets customer needs.

**Keywords:** additive manufacturing; IoT; welding





## 1. Introduction

With the advent of mechanized production methods in the latter half of the 18th century, the era of the industrial revolution began. The rigorous research works and application of scientific trends led to several technological advancements and ran towards the optimization of things. Industry 4.0 is the new name for the 4th industrial revolution, which is projected to bring significant improvements to industrial sectors. It integrates physical and digital technologies, such as analytics, robotics, AM, artificial intelligence (AI), sophisticated materials, natural language processing, high-performance computing, argument reality, and cognitive technologies. Thus, in an Industry 4.0 factory, machines are connected as a collaborative community. Such evolution helps to utilize advanced prediction tools, integrating the IoT with various manufacturing processes, so that data can be systematically processed into information to explain uncertainties, which can thereby support more updated decisions [1]. The IoT is the foundational technology that is accountable for the idea of Industry 4.0. This technology enables the logical linking of all the equipment and methods associated with the production environment, sensors, production

cells, transmitters, computers, the production planning system, strategic industry guidelines, government information, and climate, and everything is recorded and analyzed in a database [1,2]. The Internet of Things is a commonly used name for a set of technologies, systems, and design concepts linked with the coming wave of internet-connected physical objects.

Machine-to-machine (M2M) communication connects sensors and other devices to information and communication technology (ICT) networks via wired or wireless connections. IoT is different from M2M in that it also includes connecting these systems and sensors to the wider internet and using general internet technologies [3]. In the long run, it is expected that an IoT ecosystem will develop that is similar to the Internet we have today. This will allow things and real-world objects to connect, talk to each other, and interact with each other in the same way that people do today on the web. A deeper understanding of the complexity of the systems in question, economies of scale, and strategies to assure interoperability will lead to the wider acceptance and implementation of IoT solutions if combined with essential business motivations and governance frameworks throughout value chains. One of the major and upcoming revolutions in the industry is integrating IoT with AM [4]. Today, we have been seeing several advancements in this area of integration, where practical models and several ongoing research works are processing out. Almost seamless integration has occurred between additive manufacturing and the Internet of Things, and given that they are both cutting-edge technological breakthroughs, their merger was nearly inevitable. In this part of integration, we found that electronic device prototypes, such as GPS systems and circuit boards, may be created via 3D printing. This capability is being leveraged by businesses worldwide. Israeli start-up, Nano Dimension, for instance, has devoted itself to developing a 3D printer capable of producing circuit boards for the Internet of Things [5].

AM plays a vital role in Industry 4.0 and it is an innovative manufacturing technology capable of fabricating geometries in almost all shapes even with small cellular structures inside, by adding material layer-by-layer directly from the digital data file [6]. The material used to create the solid object makes these AM devices different. It uses materials like polymers, metals, thermoplastics, ceramics, composites, sand, bio-chemicals, and also some other type of materials [7–9]. However, the use of metal alloys in AM is restricted to a few components due to various reasons like input parameters, fixed bed size, availability of filler materials with respect to applications, etc. [10]. ISO ASTM 52,900, which explains several techniques related to AM, has categorized seven core technologies of AM as Binder Jetting (BJT), Directed Energy Deposition (DED), Material Extrusion (ME), Material Jetting (MJT), Power Bed Fusion (PBF), Sheet Lamination (SHL), Vat Photo Polymerisation (VPP). There are several sub-inclusive AM processes, such as Selective Laser Sintering (SLS), Stereolithography (SLA), Fused Deposition Modelling (FDM), Laminated Object Manufacturing (LOM), Poly Jet 3D Printing (PJP), Inkjet 3D Printing (IJP), Colour Jet-Printing (CJP), Three-Dimensional Printing (3DP), Multi-Jet-Printing (MJP), Electron Beam Melting (EBM), and Laser Metal Deposition (LMD), that have been heavily researched and developed, and put into commercial use by industries [11–19]. With the advancement of all these technologies, AM is evolving as the disruptive technology in the field of smart manufacturing systems and also plays a vital part in providing extensive contributions to Industry 4.0. AM offers design innovation, material waste reduction, part flexibility, inventory stock reduction, energy savings, customization, reduction of errors, and creative freedom without the cost and time constraints of traditional manufacturing [20]. However, there are some areas, such as limited material selection, part size restrictions, part cost, post-processing, dimensional control, and no custom alloying, that restrict its full-scale implementations, which have to be overcome by integrating other technologies or by following several research works [21].

Every gadget/machinery has internet access, making it easier to manage from anywhere on the Internet, with or without passwords. IoT Beacon technology allows us to remotely control and monitor pricey equipment with secure passwords. Beacon is hardware that combines Bluetooth Low Energy standard and a low-power battery to communicate

messages and information directly to a smartphone, tablet, device, or other equipment to monitor persons around the machine [1,22]. This makes AM technology more flexible and easier to process.

The International Telecommunication Union recommends that the network architecture of IoT consists of the sensing layer, the access layer, the network layer, the middleware layer, and the application layers [23,24]. These layers feature in capturing, transferring, integrating, managing, and controlling real-time network information for intelligent integrated processing. IoT possesses the capacity to assist devices in sensing the environment in which they are located, and by doing so, it has made it feasible to efficiently regulate and oversee industrial activities remotely [25,26]. The network architecture and overview of various applications of IoT are shown in Figure 1.

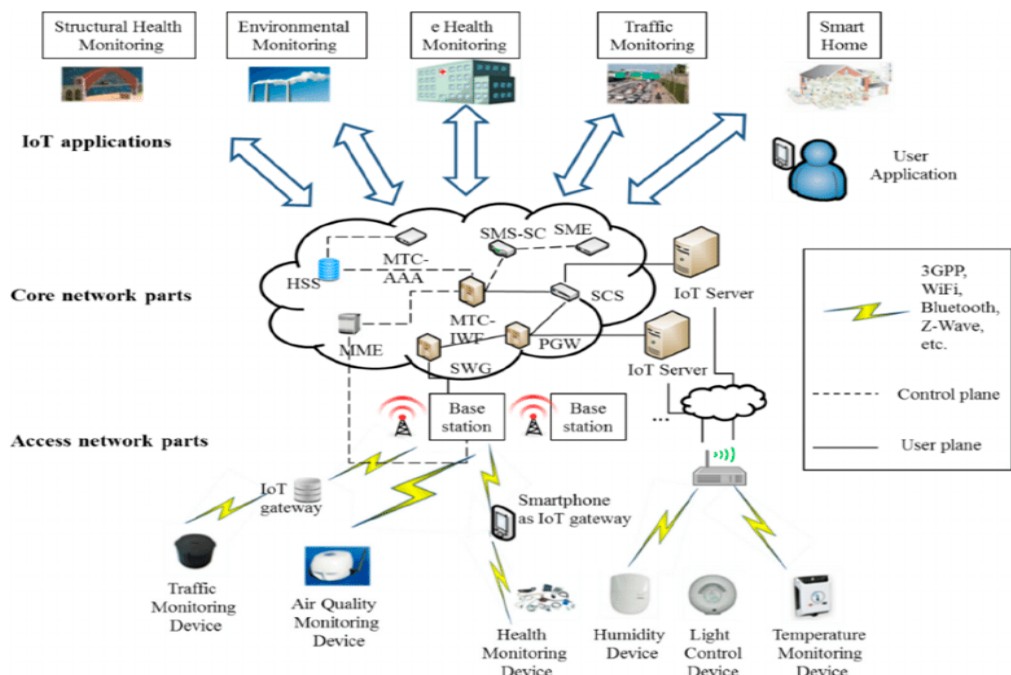

**Figure 1.** The network architecture of IoT-enabled applications [27].

We have seen that the IoT devices can be created to be specific to almost every industry and vertical organization, including healthcare, industrial automation, smart homes and buildings, and automotive and wearable technologies, which we are naming as trendy gadgets and habituating towards the comforts offered by their usage. But the thirst for technological advancements that lead to cloning or integrating IoT with other processes is happening very rapidly in several sectors. In this list, the influence of AM technology on the conceptualization of future Internet of Things breakthroughs has been enormous. The necessity for an accurate and realistic prototype is a crucial aspect of new product development. Traditional means of obtaining a prototype that fits these requirements are rather costly. With additive manufacturing, however, these advancements may be produced at a fraction of the cost and in a shorter period than with conventional methods, while also being more exact. As a result, 3D printing is becoming an integral part of producing new goods for the Internet of Things.

Internet of Things-enabled AM assists in sustainable manufacturing and leads to the evolution of industrial production systems which brings some expected benefits. It also eliminates some of the problems related to integration and data handling that are countered in AM processes. This creates an environment that is networked and comprised of "networks of networks" for the real-time management of production systems. These are the systems in which products are created, manufactured, and delivered. The Internet of Things encourages more individualization, the most efficient use of materials, and

speedy manufacture using AM processes [14,28]. When IoT devices are integrated into additive manufacturing machines, a decentralized production environment is created. In this environment, data may be sent in real time between the maker and the end-user. The final product is built according to the demands of the client, and if the customer wishes to make any modifications to the parameters of the design, they may update the data stored in the cloud. The modifications are retrieved by the AM machine, and then they are applied to the product before it is created. This results in more efficient use of raw materials and a reduction in the overall amount of time spent on the production process [14]. Thus, making it a flexible approach for rapid prototyping and creating a greater impact on the whole production scenario by integrating various levels of manufacturing. This trend creates smart and connected manufacturing which is aimed at customization, efficiency, performance, increased innovation, reduction of errors, and better management and safety.

In recent years, there have been several improvements in the manufacturing industry, including IoT, 3D printing, and digitalization. Additive manufacturing is one of the fastest-growing technologies to apply for the production of various components. However, there is a lack of skilled engineers to deal with this process. In order to solve this problem, with limited AM experts' production can be continued using IoT. However, there is no collective data and review material to understand the potential of the IoT and its applications. Hence, the present study, a critical review, was conducted to examine some of the recent trends and advancements of IoT-integrated AM technologies that exhibit a higher scope of future ruling applications. The following sections also explain the structural operations and their emerging applications in various sectors, giving both the advantages and limitations, which indicate that for tomorrow's smart factories, AM with integrated IoT has potential.

## 2. IOT-Enabled 3D Printing Applications

The implementation of Industry 4.0 has revitalized industrial processes. Smart manufacturing, in which different technologies are combined and incorporated into traditional manufacturing processes, is one of the most discussed IoT developments, with manufacturers seeking chances to establish smart factories in order to expand production processes [29]. The IoT and its associated services enable the creation of networks that incorporate the whole production process, therefore, transforming factories into smart environments; for example, industrial IoT applications are depicted in Figure 2.

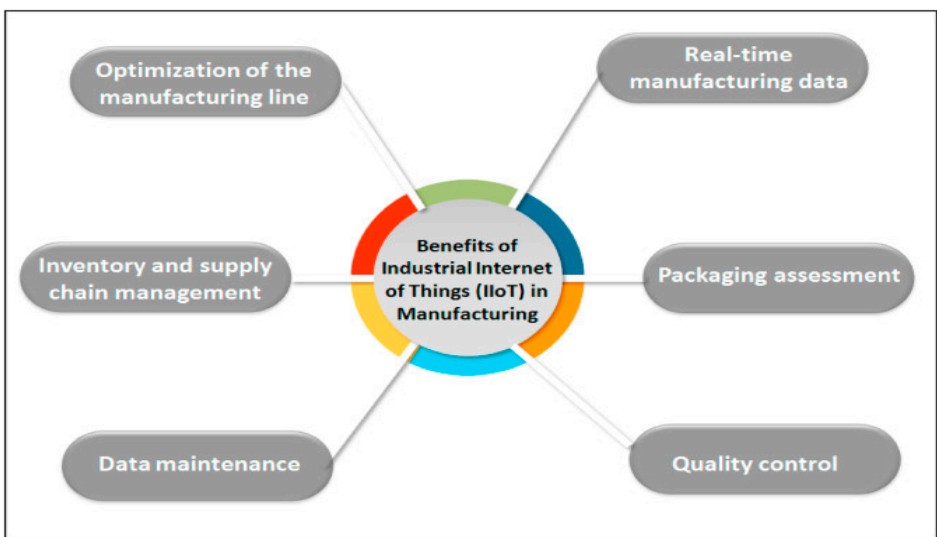

**Figure 2.** Applications of IoT in manufacturing [30].

The potential of IoT-enabled AM processes has been determined by identifying its important application areas as a consequence of an intensive analysis of the theoretical literature. While AM is a cutting-edge technology, it is making its way into many industries

by clashing with IoT technology as a result of its numerous advantages [31]. This study subject was selected due to its potential applicability in a vast array of disciplines; some of the current application trends for this technology are provided in Figure 3.

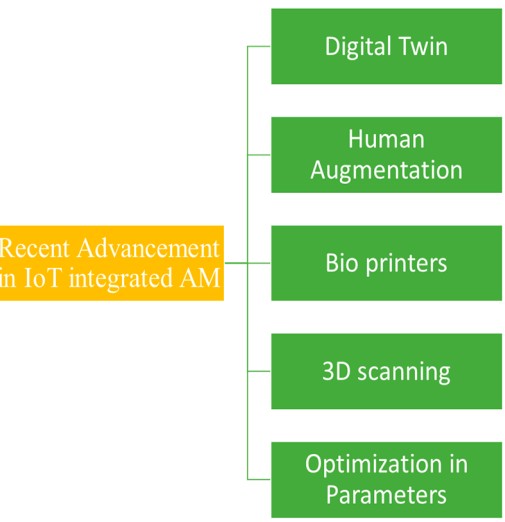

**Figure 3.** Recent advancements in IoT.

## 2.1. Digital Twin

A digital twin (DT) is a virtual version of a physical item or process that is able to collect data from the real world in order to accurately depict, validate, and mimic the behavior of its physical twin in the present and the future. The ability to make decisions based on data, monitor complicated systems, validate and simulate products, and manage object lifecycles are all made significantly easier by its presence [32]. NASA was the first association to forge the definition of DT; it has been described as "an integrated multi-physics, multi-scale, probabilistic simulation of a vehicle or a system that uses the best available physical models, sensor updates, fleet history, etc., to mirror the life of its flying twin" [33,34].

Even though digital twin technology is considered to be a recent innovation in Industry 4.0, the concept behind it has been around for quite some time. The concept of digital twin technology was first discussed publicly in 1991, but it was not until 2002 that Michael Grieves of the University of Michigan developed the idea in connection to product life cycle management (PLM) [33,35]. The model that was presented is known as the "Mirrored Spaces Model," and it is comprised of three components: actual space, virtual space, and a connecting mechanism for the flow of data and information between the two [33,36]. A similar idea developed by David Gelernter in 1991 has since become popular; in the so-named "Mirror Worlds," software models imitate reality based on information that is input from the actual world, [37]. The conceptual model that was suggested by Grieves in 2006 was originally referred to as the "Mirrored Spaces Model," but in 2006, it was renamed the "Information Mirroring Model" [33,35,38].

However, at the time, there were no practical implementations of digital twin due to a lack of technological developments in simulation software, data management, and storage systems, limited or no connection of devices with the Internet, immature machine algorithms, and other issues. In the end, John Vickers of NASA came up with a brand new phrase, which was "digital twin," in the year 2010. In the NASA roadmap, it was also referred to as the "Virtual Digital Fleet Leader." Figure 4 outlines the timeline and development of digital twin throughout the course of human history.

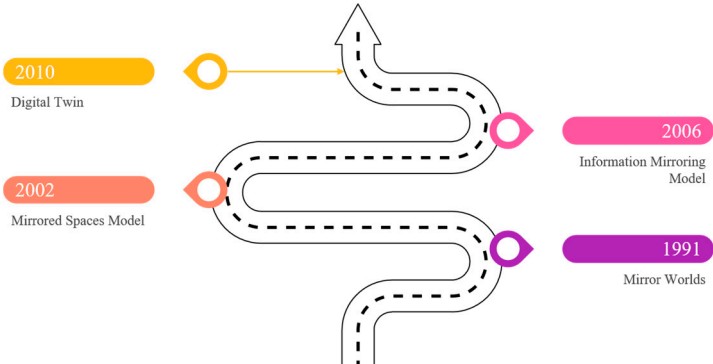

**Figure 4.** Digital twin evolution timeline.

Digital twins (DTs) are formed by employing technology that merges the IoT, software analytics, AI, and specific network visuals to imitate real assets or processes. DTs are used to model and simulate complex systems (physical twins). In DT technology, we embed connectivity between physical assets and their digital counterparts and enhance their connectivity with their digital representations using six primary features, such as sensors, actuators, data, integration, analytics, and the digital twin, as shown in Figure 5. In addition, we embed connectivity between physical assets and their digital counterparts in DT technology. As the Internet of Things implementations provide access to massive data and enormous digital ecosystems, it is also becoming simpler to construct and maintain digital twins with a high level of realism. In the years to come, digital twin technology is going to establish itself as one of the most important software tools for changing product creation.

According to Shimamura et al. and Gaikwad et al., the DT has the potential to considerably enhance the AM process by optimizing the various process parameters and providing a framework for the design of the component and support structure [39,40]. It can also, monitor possible process failures with simulations and real-time data from sensors, and store and process data obtained from a variety of sensors [40].

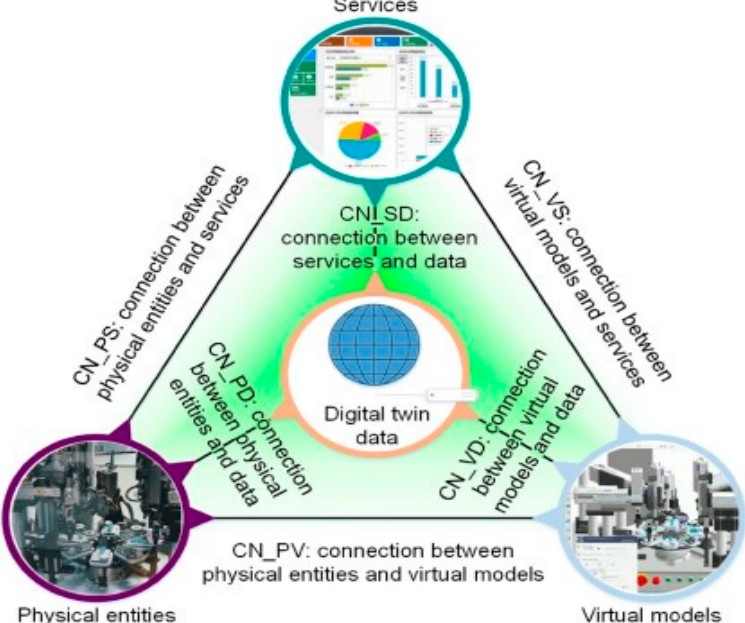

**Figure 5.** The working of the digital twin model [41].

The elimination of printing errors is the single most important task that can be assigned to a DT by an AM since this will result in more efficient use of resources, which in turn will raise a company's profit and lower its expenses [39]. This DT feature will assist in

improving the flexibility, design, printing quality, and printing style of contemporary 3D printing. Sensors and other features of DT are used to acquire data from the physical 3D component so that it may be integrated into the system. As a result, the IoT-enabled 3D printing industry is finding its place by enhancing the capabilities of contemporary AM procedures. Because of this connectivity, the sensors are now able to send the data that they have collected to a user. The data that is acquired is of the utmost importance when it comes to determining and fixing any issues that may arise throughout the process of 3D printing.

There will be many digital twins representing various physical realities in the coming years, such as people, processes, places, and objects. This technology is driving optimal performance and innovation in a variety of industries. DTs are now changing their wings from lab experiments to practical applications where we can see that the United States Air Force (USAF) is working on Reengineering Aircraft Structural Life Prediction using a digital twin. The proposed model utilizes DT in integrating the computation of structural deflections and temperatures in response to flight conditions for developing safety factors in maintenance and operations. This shows the importance of DTs in the occupancy of the 3D printing industrial world. As a relatively new form of technology, its application is becoming more common in a variety of settings, such as the industrial sector, the automotive industry, the medical field, smart cities, and other areas, as shown in Figure 6.

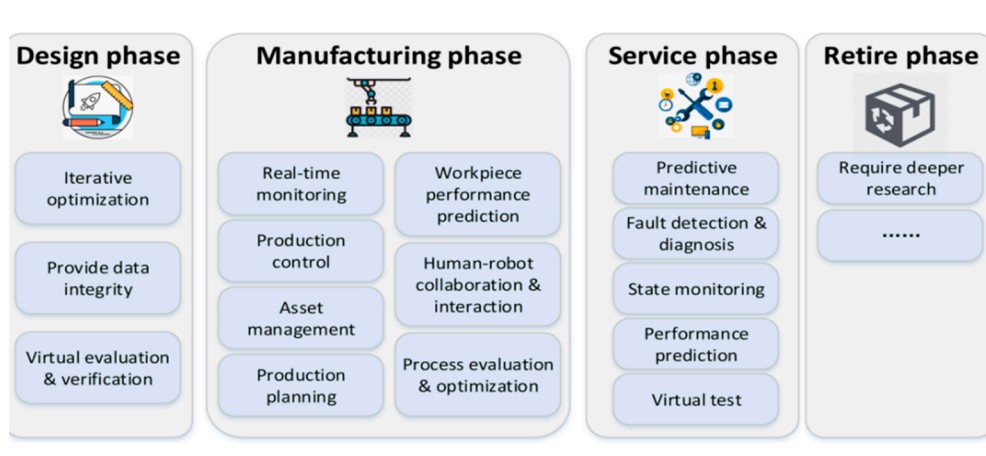

**Figure 6.** Applications of digital twin [42].

Additive manufacturing applications, especially in metallic materials, are still facing several limitations before becoming fully functional in several manufacturing processes; however, much research is being performed for implementing IoT-integrated AM in DT and will be the solution to these problems. A study on the feasibility of DT in AM indicates that the creation of a data centralization tool, which is installed when needed and acts as the foundation of the DT to enable automatic data flow between different installed sensors and allow the generation of decisions for controlling several processes with the help of controllers, makes it fully viable as a widespread manufacturing process. DT excels at helping streamline process efficiency, as we find in industrial environments with co-functioning machine systems in additive manufacturing, as shown in Figure 7.

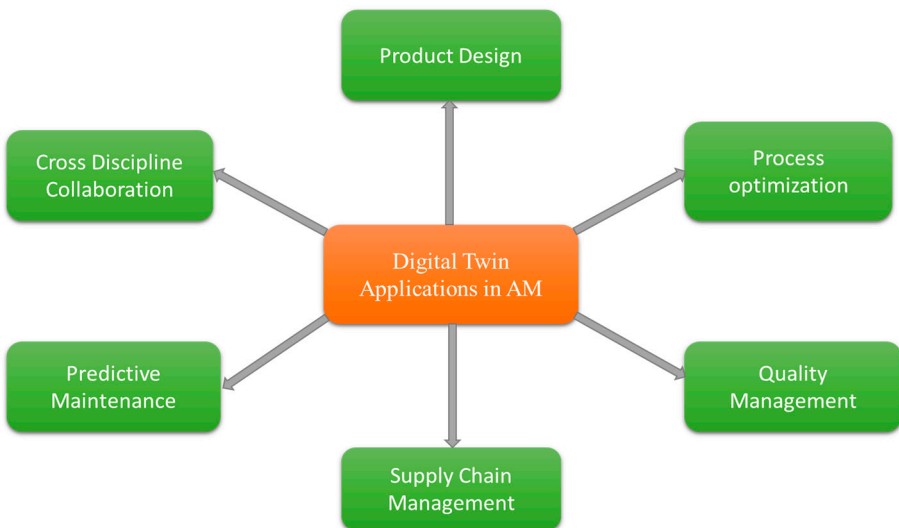

**Figure 7.** Digital twin applications in AM.

Although there are other traditional systems, such as drafting and modeling software, analysis, and control strategies, there is higher scope for the automation and optimization of twinning physical models in order to encounter problems through DT (see Figure 7 above). A digital twin, which involves building a bridge between the physical and virtual worlds of printing, can cut down on the number of trial-and-error tests, reduce the risk of defects, shorten the amount of time that elapses between the design stage and the production stage, and make printing more metallic products more cost-effective [43]. The Cyber-Physical Production System (CAPP) acts as a platform for integrating several virtual twins into the physical systems [44,45]. DT helps additive manufacturing in terms of product design, process optimization, supply chain management, quality management, cross-discipline collaboration, predictive maintenance, and analyzing the customer experience. Thus, implementing the IoT in additive manufacturing will help the process to consume less power, ensure a safe manufacturing environment, increase the rate of production, allow for mass production and long-term viability, reduce material wastage, and make the process more cost-effective.

## 2.2. Human Augmentation

In medical terms, augmentation means adding (augmenting) a medication or other treatment to improve outcomes. Human augmentation is the capacity to conduct physical or mental acts with the aid of instruments that practically merge with our bodies, hence expanding our capacities [46]. Human augmentation refers to the addition or enhancement of biological functions by technological techniques such as surgery, implants, skin grafts, or tissue expansion with the consideration of compensation and augmentation as illustrated in Figure 8 [47].

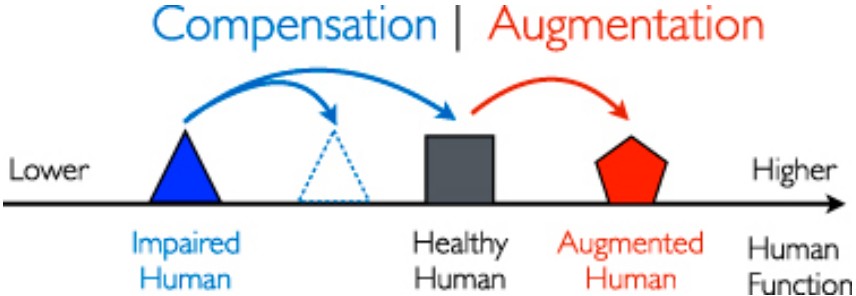

**Figure 8.** Human augmentation [47].

Human augmentation has been practiced ever since the beginning of humankind. For a significant portion of human history, the concept of augmenting the human body has been a topic that has been explored in both fiction and scientific accomplishments [48,49]. In the past, as our machines have become more advanced, our scientists have concentrated more on the machines themselves than they have on humans, but this is starting to change. Recent developments in the life sciences have resulted in the establishment of a new subject known as HA, which is an interdisciplinary discipline that has the potential to disrupt every aspect of our existence [50]. The essay "Human Augmentation" by Douglas Engelbart, which was written in 1962 and published by the Stanford Research Institute, is widely considered to be one of the most influential works on the subject. The study establishes the groundwork for the idea of enhancing the human mind [46].

Human augmentation refers to technologies that increase human productivity. It also improves or restores the human body or mental capabilities. Its goal is to improve the human experience in both cognitive and physical ways [51]. HA is also proposed as the application of science and technology to increase human performance, either temporarily or permanently. This area may be broken down even further into human performance and human augmentation, both of which are depicted in Figure 9. It is explained that human performance optimization is the application of science and technology up to the biological potential without adding new capabilities, whereas human performance enhancement is the application of science and technology beyond the limit of biological potential and has the potential to add new capabilities beyond those that are innate to humans [50]. People that participate in human augmentation have the chance to increase their creative output, their productivity, and the societal effect they have [47].

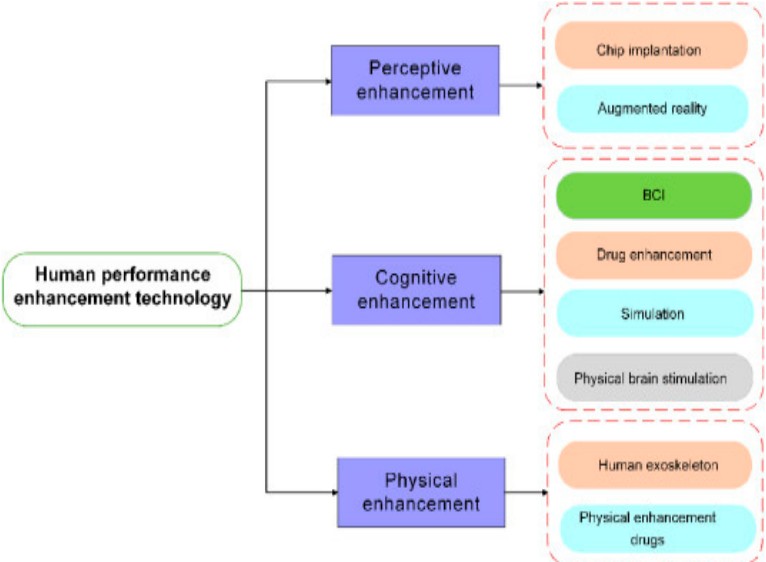

**Figure 9.** The delineation between human performance optimization and human performance enhancement is outcome-based [50].

HA affords individuals the chance to be more inventive, productive, and influential in society [47]. It is an interdisciplinary area that discusses and/or manages cognitive abilities. This is accomplished through the implementation of sensing and propulsion technologies, information mixing and separation approaches, and AI-based procedures [48]. The technological advancement in applications of the Internet of Things in the field of additive manufacturing has made HA a very trendy and future-centric technology where development in this field can set the next generation human benchmark in the evolution of humans themselves [47]. These advancements in technology will improve critical areas such as human health, quality of life and performance, and our ability to operate and perceive our surroundings [51].

Additive bio-manufacturing is defined as the utilization of 3D printing for medical or non-therapeutic "human enhancement" reasons, regardless of whether they include the synthesis of biological material. It covers any application that rehabilitates, supports, or enhances the biological functionality of an organism [52]. There are many different technological paradigms and user interface (UI) approaches that have been developed with the intention of making the interaction feel more intuitive and effective. The system is able to receive data from a variety of sensors and give the user information in real-time using a variety of modalities, such as visual, aural, and haptic presentations [51]. HA technology has also been made evident by recent breakthroughs in other fields of technology, such as the interface of a brain-computer or brain machine, high-resolution virtual environments (VEs), and algorithms for optimization [48,53–58].

The AM market is now attracting HA media because it can let us become "super-humans," or "Human 2.0" in a more flexible way than any other existing method. With advances in brain-computer interfaces, we are also approaching augmented human intelligence using AM. But not all forms of HA technology provide quality outcomes and the same is applied in AM fields, where more research is required to improve AM applications in HA. Innovation in HA will occur on three action fonts, as explained in Figure 10.

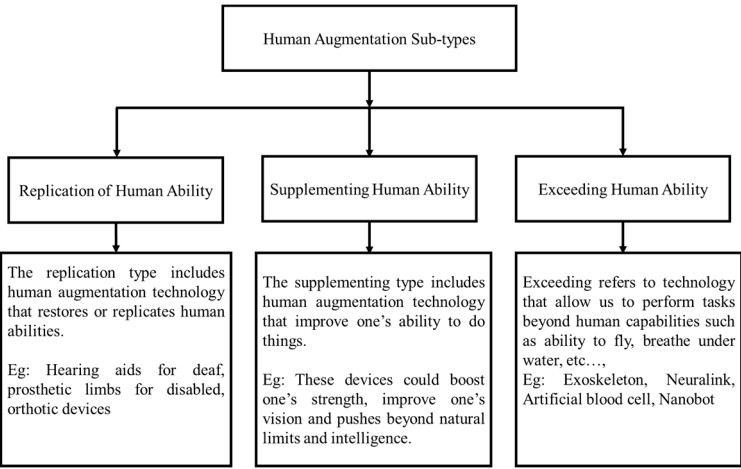

**Figure 10.** Sub-types and innovation action points of HA.

Creating cognitive and physical advancements as an essential component of the human body is the primary emphasis of the area of HA, which is also frequently referred to as "Human 2.0." One illustration of this would be the application of active control systems in the manufacture of limb prostheses with features that are capable of outperforming the greatest level of natural human performance [59]. The use of external equipment, such as eyeglasses, binoculars, microscopes, or very sensitive microphones, has also been shown to result in the development of enhanced capabilities. In recent years, technology such as augmented reality and multimodal interaction has made it possible to enhance people in a way that is not intrusive. In the future, technology will further decide the types of humans [60]. Some real-life applications of human augmentation in AM are utilizing smart glasses and contact lenses, naked prosthetics, Neuralink, human genetic engineering, neurotechnology, Cyberware, strategies for engineered negligible senescence, 3D bioprinting, and nanomedicine. Different forms of human enhancement technologies are either on their way or are currently being tested, trialed, and implemented for 3D printing applications. A few hypothetical human enhancement technologies are under speculation, such as mind-uploading applications, exocortex, and endogenous artificial nutrition [61].

With the help of HA, the AM process is finding some advancements in its printing strategies and some of the intriguing pieces of human augmentation in AM are visualizing, Sensors Augmentation, Brain-Computer Interfacing, Natural Language Generating,

Real-Time Conversation Translation, Modern Hearing Aid with iPhone Compatibility, Exoskeletons for Workplace Safety, Neurally-Controlled Artificial Limbs, and Superhuman Sensory Enhancement [62]. We have seen that humans have adopted computers for several technological advancements in order to address problems in various sectors; although in the future, there is a high scope for computers adopting humans towards the enhancement of capabilities beyond the potential limits.

### 2.3. 3D Bioprinters

The use of additive manufacturing in biological matter formation, such as cells, tissues, blood cells, etc. is called 3D bioprinting. Bioprinters operate virtually similarly to 3D printers, with one key distinction. Rather than delivering items such as plastic, ceramic, metal, or food, they deposit layers of biomaterial, which may contain living cells, to construct complex structures such as blood arteries and skin tissue. Currently, the majority of 3D bioprinting materials are derived from either natural or synthetic biomaterials. Bioactivity is the primary benefit of organically derived materials (such as collagen, chitosan, gelatin, hyaluronic acid, alginate, and fibrin). They often have a high degree of resemblance to the extracellular matrix (ECM) and are biocompatible. The IoT helps to avoid the delay in producing the organs. If we fully use IoT technology in bioprinting, then we can decrease deaths due to organ damage in accidents. For example, if a person's heart is partially damaged then with the help of IoT sensors and systems we can understand the complete requirements of the damaged heart, then those paraments can be given as the input to the bioprinters so the new synthetic heart is made before the damaged heart fails completely. However, natural biomaterials are frequently poor in terms of their mechanical qualities, even after cross-linking. Synthetic materials (such as PCL, PEG, PLA, PEEK, and Pluronic) have better advantages over natural materials because they can be designed with specified physical qualities and have more uniformity [63]. For use in medical treatments, training, and testing, bioprinting may generate living tissue, bone, blood arteries, and potentially complete organs. Cell development and several bioprinting methods are depicted in Figure 11.

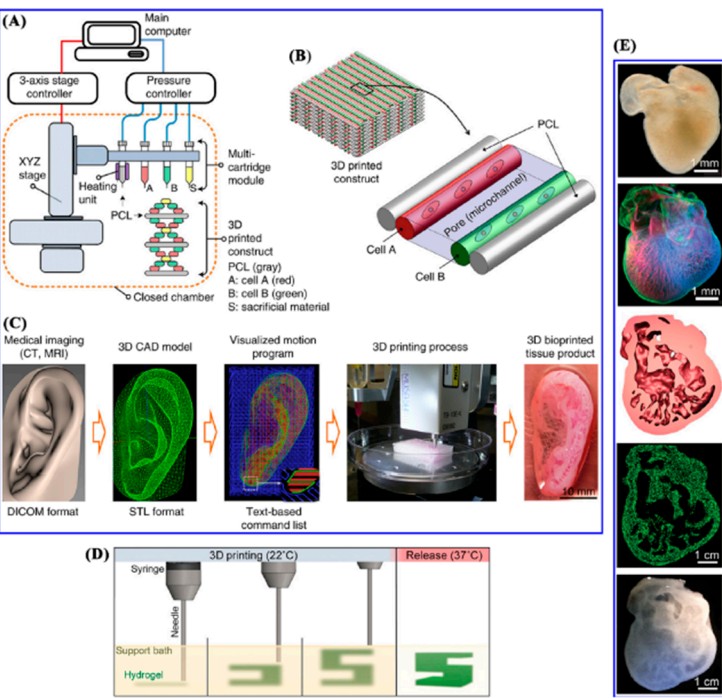

**Figure 11.** ITOP systems with various bioprinters (**A**) process flow chart (**B**) 3D printed object (**C**) 3D printing process using CAD model and (**D**) 3D printing bath with syringe head (**E**) quality assurance of the 3D printed organs [64].

In the early stages of healthcare, the loss or failure of organs and tissues is a challenging and expensive issue. Worldwide organ shortages have prompted research on tissue engineering, specifically the building of a cell-scaffold microenvironment to stimulate the regeneration of diverse types of tissue, such as skin, bone, cartilage, tendon, and cardiac tissue [65]. Currently, it is not possible to bioprint completely functional organs for transplantation in three dimensions. Figure 12 illustrates the development of bioprinting. However, it is undeniable that bioprinting processes have developed substantially. Several pioneers, including Vladimir Mironov, Gabor Forgacs, and Thomas Boland, anticipated decades ago that the natural combination of technologies, including cell patterning and commercial inkjet printing, for the construction of living structures may one day be used in human organ transplantation [66].

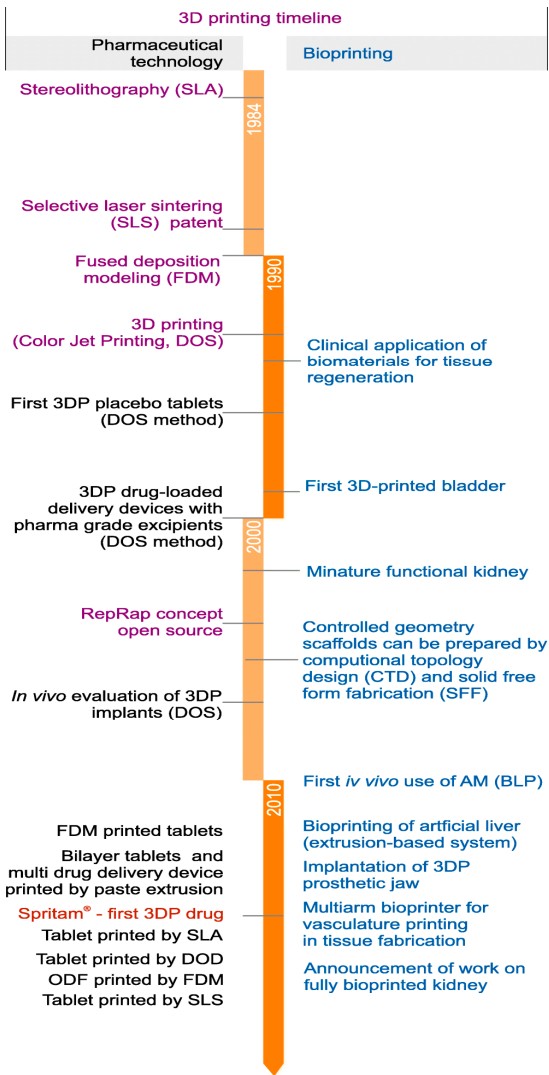

**Figure 12.** Bioprinting evolution timeline [67].

Year 1984 marked the start of 3D printing when Charles Hull pioneered stereolithography (SLA) for printing 3D applications from digital data. Bioprinting was shown for the first time in 1988 when Klebe utilized a conventional Hewlett-Packard (HP) inkjet printer to deposit cells using cytoscribing technology [68]. Forgacs and colleagues [69] concluded in 1996 that apparent tissue surface tension was the macroscopic expression of molecular adhesion between cells and gave a quantitative measure of tissue cohesion. Odde and Renn exploited laser-assisted bioprinting to deposit living cells for the development of mimics with complicated anatomy for the first time in 1999 [70]. The year 2001 saw the direct

printing of a bladder-shaped scaffold and the implantation of human cells [71]. In 2002, Landers et al. revealed the first extrusion-based bioprinting method, which was eventually commercialized as the "3D-Bioplotter" [72]. In 2003, Wilson et al. created the first inkjet bioprinter by adapting a conventional HP inkjet printer [73]. Their team accomplished cell-loaded bioprinting using a commercial SLA printer a year later [74]. The same year saw the development of 3D tissue containing only cells (no scaffold). The year 2006 saw the use of electrohydrodynamic jetting for the deposition of live cells [75]. Norotte et al. produced scaffold-free vascular tissue by bioprinting in 2009 [76]. In 2012, Skardal et al. performed in situ bioprinting on mouse models [77]. In the years that followed, several innovative bioprinting items were introduced, including articular cartilage and an artificial liver in 2012, and tissue integration with the circulatory system in 2014, etc. [78,79]. Gao et al. employed a coaxial technique for the manufacturing of tubular structures in 2015 [80]. In 2016, Pyo et al. implemented DLP-based quick continuous optical 3D printing [81]. The same year, the research group of Anthony Atala produced a cartilage model utilizing an integrated tissue-organ printer (ITOP) [81]. Noor et al. succeeded in producing a superior scaled-down heart in 2019 [82]. A few months later, Kang et al. [83] accomplished the bioprinting of collagen human hearts at various sizes using the freeform reversible embedding of suspended hydrogels (FRESH) method [66].

Bioprinting technology has advanced significantly, yet it faces challenges such as insufficient biocompatibility, hazardous breakdown products, and a dearth of bioactive ligands [63]. The challenges can be solved by the Internet of things (IoT) using sensors. The IoT monitors the entire bioprinting process with accuracy and repeatability. If the entire bioprinter environment is integrated with the IoT then the sensors in the IoT will take care of all the parameters throughout the process. That is, first the IoT will find the best method for the tissues from the different bioprinting technologies. Then the IoT will select the material and parameters for the printing process. If any toxic waste is produced during the bioprinting process, it is separated and stored properly by the sensors of the IoT. The IoT will control the biomaterial deposition so there is good biocompatibility between the cells of the tissues. The IoT also integrates other technologies, such as 3D scanning in bioprinting. If an organ is damaged in an accident or for any other reason, then that organ should be replaced with a new organ with the same tropology and geometry within a limited time; in this case, the IoT can use 3D scanning technology to get information on the topology and geometry from the damaged organ.

Future research activities will concentrate on the creation of innovative 3D bioprinting materials with superior biocompatibility, enhanced printing capability, and acceptable mechanical qualities. Various uses of 3D printing include the study of cellular behavior, tissue pharmacodynamics, and toxicological pathways, to name a few. Despite the excellent accuracy and repeatability of 3D printers, the printing of organs and functioning tissues with complete architectures still requires assembly layer-by-layer with "bioglue." Bionics with sufficient biocompatibility and mechanical strength that can be employed to produce biological functions are the primary technological obstacles. For soft and hard tissue engineering, hydrogels and ceramics have been utilized, respectively. In the meanwhile, personalized 3D printing technology will result in a number of regulatory obstacles pertaining to printed product oversight. Nevertheless, it is of the utmost importance that management establishes and improves the appropriate rules and regulations in order to ensure the continued and sustainable growth of 3D printing technology. Printing micro-organs, such as pancreas islet tissues that operate in the absence of the whole pancreas structure, will likely make significant development in the near future, which will assist hundreds of millions of diabetes people worldwide. Such changes have successfully resulted in the manufacture of micro-liver samples for drug metabolism testing.

### 2.4. 3D Scanners

3D scanning is the process of studying an object or environment in the real world to acquire data about its shape and even its appearance (e.g., color). The collected infor-

mation may subsequently be utilized to create digital 3D models. Dark, reflective, glossy or translucent objects can provide several challenges for 3D scanners. Utilizing 3D scanning technology, a physical model is converted into a digital 3D computer-aided design (CAD) file. This digital output is utilized effectively for the design and fabrication of bespoke components using AM technology. Figure 13 illustrates the distinction between conventional manufacturing and 3D scanning. Apart from the social sectors, 3D scanning supports a range of conventional technical industries. Figure 14 depicts the widespread potential applications of 3D scanning, including scientific and education sectors, design and manufacturing domains, reverse engineering professions, art and design industries, etc. Additionally, technology provides various benefits. Specifically, enhanced workflow processes, correct resolutions, an automated system as a whole, quality services, research views, etc., make this instrument a more particular quality service for the engineering industry at present [84].

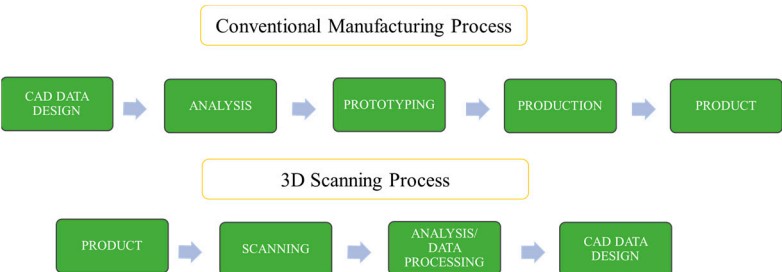

**Figure 13.** Conventional Vs 3D scanning process.

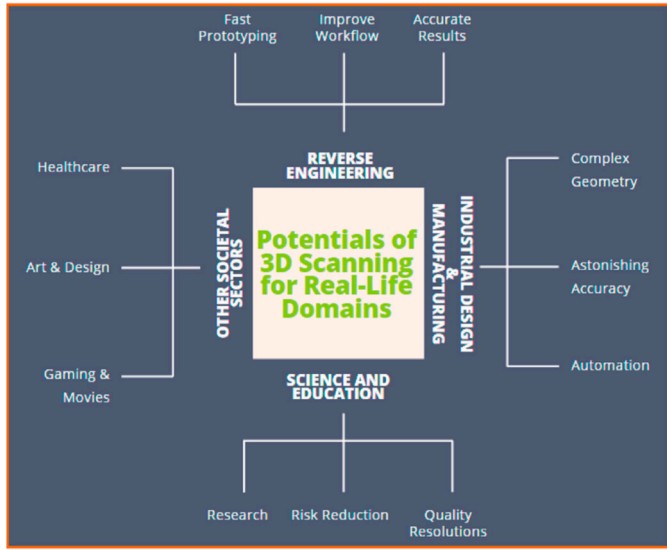

**Figure 14.** Potentials of 3D scanning for real-life domains [84].

3D scanning technologies are utilized to scan physical components that frequently lack drawings or CAD data in order to digitally copy the data. Using the scan data, quality managers may measure and compare the actual design with digital information. The accuracy of the component's dimensions may be rapidly computed and verified. Precision and construction quality analysis, including fitness checks, gaps, and alignment, is helpful for components with complicated geometry and old or broken instruments. However, the engineers are trained in the utilization of robust models scanned by this technique. This information is stored as a feature in the logical measure and specifies the necessary format of an object. Changes to the dimensions of a single element may be made in the solid CAD, and the other model updates will accommodate the modifications. Workflows utilizing scanning technology are easily extensible to reproduction and reconstruction, reverse engineering, and metrology [84].

In earlier days, the modelling of a complex part is difficult due to complex structures, curves, and irregular shapes. Due to this, the final 3D Model varies from the original product. To overcome this problem, 3D scanning technology was discovered. The first 3D scanning technology was created in the 1960s. The earlier scanner uses light, cameras, and projectors to perform the task. Due to this limitation of this equipment, it often took a lot of time and effort to scan objects accurately. After 1985, there was a replacement with scanners that used white light, lasers, and shading to create a 3D model of a given object. With this advancement, it was possible to build up highly complex models. Some of the manufacturing components required very high accuracy but modeling these components with conventional modeling software made the model somewhat inaccurate due to the complex curves. So, for these highly accurate components, we use 3D scanning technology. Scanning makes it possible to create a 3D model of an existing part or its environment. The digitized part can be used for producing a replacement part or creating a better-optimized entity in design. 3D scanning is also used for inspection purposes; that is, 3D scanners are used to inspect the final product of additive manufacturing. Figure 15 shows the 3D scanning process used in the inspection.

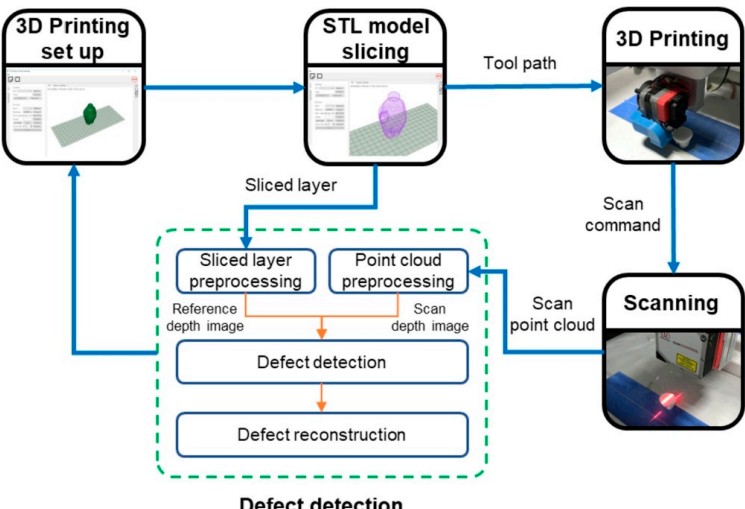

**Figure 15.** The 3D scanning technology used in inspection [85].

Therefore, additive manufacturing with 3D scanning using the Internet of Things (IoT) makes the process very easy. The IoT monitors the scanning process very accurately. It will suggest the best method of scanning for the best result and integrate the 3D scanner and 3D printer by generating the STL file. From many types of research, we have understood that IoT-integrated AM technology created a seamless scope for 3D scanning methodologies by overcoming certain difficulties such as material parameters, the topology of the body, data integration, and handling between the scanning and printing devices with the involvement of sensors, actuators, and other IoT connectivity devices. This trend involves recent advancements in technologies like aerospace, reverse engineering, surgical planning, performing architectural surveys, and so on. With much further research and technological inventions, we could achieve remarkable standards in scanning technologies.

The technology of 3D scanning has only been available to us for a very short period of time, yet its development is proceeding at a faster rate. The potential applications of 3D scanning are becoming further investigated day by day. We have discussed at length how the incorporation of AI into this technology will make the process more user-friendly and convenient. We can now assert that the future of 3D is more promising than we could have ever imagined. The use of AI will complete the process of 3D scanning, making it as simple as recording a movie with a smartphone. Scanning in three dimensions is expected to grow rapidly in the next years, and many sectors will soon require the service.

### 2.5. Input Parameters Optimization

Additive manufacturing enables layer-by-layer deposition to produce three-dimensional things mainly depending on process parameters. Irrespective of the applications, the materials, shape, and quantity of the 3D printing parameters decide the quality of deposited output and thereby decide the material's properties. Wire arc additive manufacturing (WAAM) generates all AM techniques' highest deposition rate and arc energy [86,87]. Hence optimizing and predicting the parameters is essential and offers certain advantages, such as being cost-effective, and time-efficient, and solving complicated designs, especially for large-scale components. In addition, it is crucial to minimize the material waste that enables lean production today. Some of the major developments after interconnecting the IoT with AM process parameters are process selection, material smart selection, post-processing, non-destructive testing applications, etc.

Each metal additively manufactured object has its own set of specifications. Choosing process parameters to match these product specifications is made difficult by the vast number of process parameters available. Moreover, to achieve objectives such as geometrical requirements, good surface finish, improved mechanical properties, processing speed, etc., parameter optimization is required. In addition, the empirical method is commonly used in the industrial sector to reduce experimental effort [86]. A comprehensive investigation of the bead geometry constituting the layers is required to predict improved dimensional control of components. In order to better appreciate optimal geometry, researchers have employed various modeling and optimization techniques to merge many goals functions into a single objective function [88,89]. In order to acquire the ideal set of parameters, the combined purpose of utilizing grey-relational analysis (GRA) is therefore crucial. Researchers nowadays implement modeling using AI techniques. For instance, as shown in Figure 16, wire arc additive manufacturing has various phases in its process. When the IoT is coupled with the WAAM process, step process accuracy will be possible. Some researchers, such as Chigilipalli et al., have predicted the shape of deposited weld beads using neuro-fuzzy modeling of the machine learning technique and have produced high-quality, multi-layered WAAM deposits [90]. This study shows the importance of the IoT in any AM process.

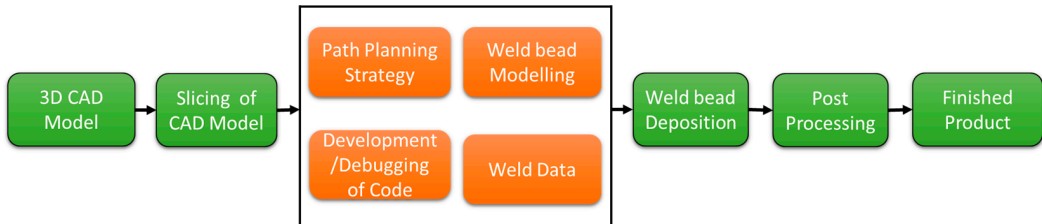

**Figure 16.** WAAM process phases.

The IoT helps AM by monitoring product quality and identifying defects that help to improve the 3D printing process [91]. IoT-based additive manufacturing helps in a defect detection system, for instance, by providing novel insights into AM processes by identifying implicit knowledge and establishing a link between input parameters and output quality to ensure defect-free printing [92]. IoT-based machine learning algorithms help solve such types of challenges effectively. Zhu et al. [93] argue that such type of defect detection system for laser-based AM processes aids in ensuring product quality and improving printing process efficiency. Moreover, anticipating deposition process parameters improves deposition quality. Manjunath et al. have identified that the output expectations of deposition depend on the input training data [94]. In addition, Cheepu has investigated such defect detection and prediction modeling strategies that aid in the prediction of defects during 3D printing [95]. Using the IoT, the data from the defect analysis of actual deposition plays a key part; thus, thorough observations and correct input recommendations are required.

Studying the functionality of materials that are new to the AM field also requires lots of research before going into industrial usage. At this stage also, the IoT plays a crucial place in solving complexity, including multi-material builds, controlling the cost of the build material, smart composite materials, cost of material, cost of post-process optimization, quality, microstructure control, and material build quality. Apart from these, some of the manual activities such as powder filling, component removal, and process interruptions, so a machine operator, is required. But when the IoT is coupled with the optimization of these processes and implemented in industry, then it will become the most economical for all the industries.

From the above study it was observed that, when additive manufacturing is connected with the Internet of Things, it becomes substantially more effective. Combining additive manufacturing with the Internet of Things yields various extra features, including remote monitoring, remote operation, and reduced processing needs. The results of the studies led us to conclude that the system may be run with less processing power and remote operation and monitoring while retaining acceptable print quality. The decrease in print quality caused by the IoT technique is minor when compared to that of a conventional desktop printer.

### 3. IoT-Based AM Applications and Limitations

IoT has been implemented in various sectors for enhancing the efficiency of the technology and to improve the production levels. However, there are several limitations along with their merits for each application as given in Table 1.

**Table 1.** IoT-enabled AM Applications, Advantages, and Limitations.

| S. No | Name | Applications | Advantages | Limitations |
|---|---|---|---|---|
| 1 | Digital Twin | Reengineering Aircraft Structural Life Prediction Cyber-Physical Production System (CPPS) Simulation of medical products Model-Based System Engineering (MBSE) Agriculture, healthcare, mining, robotics, | Speed prototyping and product redesigning<br>Real-time simulation<br>Two-way data integration and interaction<br>Predictive maintenance<br>Lifecycle management platforms | Issues related to data privacy and cyber security<br>Lack of standards, frameworks, and regulations<br>High cost of implementation<br>Required AI and big data for long-term data analysis<br>Digital twin is based on 3D CAD models and not on 2D drawings |
| 2 | Human Augmentation | Brain implants and artificial body parts implants<br>Neuralink<br>Exoskeletons for workplace safety<br>Hololens<br>Human genetic engineering | Support or improve human capabilities and performance<br>Reduce social injustice<br>Alleviate social inequalities<br>Improvement of overall physical and mental health Enhance body integrity | Blurring boundaries between advanced therapies and medical devices<br>Blurring boundaries between the biomaterial and non-biomaterial products<br>Issues of safety and informed consent<br>Ethical concerns<br>Threats of reality modification |
| 3 | 3D Scanning | Aerospace<br>Surgical planning<br>Reverse engineering<br>Conduct architectural surveys<br>Artists and art historians | Rapid production with customized design<br>The increased success rate of treatment<br>View detailed anatomy from different angles<br>Time and cost efficiency<br>Improved learning and reduced risk | It cannot scan beyond the surface level<br>It cannot determine the material being scanned<br>It cannot transparent or reflect objects<br>We cannot show the inner details<br>Precision doubts in droplet placement |

**Table 1.** *Cont.*

| S. No | Name | Applications | Advantages | Limitations |
|---|---|---|---|---|
| 4 | Bio Printing | Reconstruction of body tissues | An alternative to fill the organ shortage | Emission of harmful chemicals |
| | | Transplantable organs Skin | Fast than organ donation Decrease animal killing | Poor biocompatibility Lack of bioactive material |
| | | Muscular-skeletal tissue | The accurate shape and size of the organ can be printed | Need of low viscosity biomaterial |
| | | Myocardial tissue Blood vessel | | |
| 5 | Optimization in parameters | Deposition/scanning speed | Reduction in production cost | Temperature controlling during deposition |
| | | Design of AM product | Reduction in material wastage | Surface roughness issue |
| | | Path planning | Controlled deposition | |

## 4. Conclusions

This literature-based analysis focused on the importance of integrating the IoT and AM technologies. This review will give engineers and future researchers in academia and industry applications a better grasp of how to attain the IoT and state-of-the-art AM. AM is a dominant trend in manufacturing technology; however, it cannot replace conventional manufacturing techniques due to its inability to mass produce quickly. As a viable solution, this article suggests that IoT adds benefits to AM by enabling improved control and management of many processes through the incorporation of new tools and technology. Collecting data from processes, machines, etc., and providing them to decision-makers on a real-time basis enables the on-demand manufacturing of customized products. Conventional processes can deliberately be replaced by intelligent technologies (IT), limiting human intervention, increasing productivity, developing design standards, etc. This study compared the merits of implementing this technology with its limitations and some of its most recent applications.

1. Human augmentation can radically transform the community for better or worse by extending human capacities beyond their natural boundaries with the help of rapid prototyping of several augmenting devices. It also plays a significant role in enhancing human productivity through new emerging concepts of AI.
2. Digital twin technology will be a future hologram of several systems by integrating its sensory technologies for replicating physical twins with modern manufacturing 3D printing technology, and this can overcome several drawbacks in the manufacturing field by decreasing complexity in designing and prototyping.
3. The performance of 3D scanners will improve by overcoming restrictions in data collection, modeling, and inspection parameters through the integration of IT technologies and the facilitation of AM printing.
4. Bio-printers face challenges in handling and storing toxic wastes produced during the bio-printing process, and these can be minimized with this technology by close monitoring and this will lead to advancement towards a greater extent in the future.
5. The ease of printing processes using IoT-enabled AM technology is helping to overcome printing defects such as porosity, surface roughness, microstructural discontinuity, etc. The IoT is helping to avoid those defects by proper selection of parameters during 3D printing.

Based on the IoT, which connects the digital and physical worlds through AM processes, the concept of IoT-integrated AM can contribute significantly to assisting new manufacturing processes in a variety of industries, as well as promoting gains such as productivity, customization, ergonomics, flexibility, automation, integration, optimization, and innovation. Based on the challenges presented in this work, future research efforts should

focus on (1) simulation and modeling techniques to reduce computational complexities; (2) IoT data processing and analysis through big data, ML, and AI; (3) 5G communication; (4) interoperability and integration of simulation, modeling, analysis, and visualization software; (5) edge and cloud computing capabilities in advanced microprocessors, etc.

**Author Contributions:** B.K.C.—Conceptualization, methodology, software, validation, formal analysis, investigation, resources, data curation, writing—original draft preparation, writing—review and editing, visualization, supervision, project administration. T.K., G.B.—Investigation, resources, data curation, writing—original draft preparation, writing—review and editing, visualization. S.N.C.—Investigation, resources, data curation, writing—original draft preparation, writing—review and editing, visualization. R.K.K., G.B.—Investigation, resources, data curation, writing—original draft preparation, writing—review and editing, visualization. M.C.—Conceptualization, methodology, software, validation, Investigation, resources, data curation, writing—original draft preparation, writing—review and editing, visualization, project administration. All authors have read and agreed to the published version of the manuscript.

**Funding:** This research received no external funding.

**Institutional Review Board Statement:** Thanks go to the DST-funded IDEA Lab, a research center accredited by JNTU Kakinada, at Vignan's Institute of Information Technology (A), Visakhapatnam, 530049, India.

**Informed Consent Statement:** Not applicable.

**Data Availability Statement:** Not applicable.

**Acknowledgments:** The authors are grateful to Venkata Charan Kantumuchu, Global Quality Director at Electrex Inc., Hutchinson, Kansas, United States, for his time and efforts throughout the writing of this article. Your insights and in-depth debate on the technological details of IoT in welding and additive manufacturing were of great assistance to us in the preparation of the article.

**Conflicts of Interest:** The authors declare no conflict of interest.

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
