# Peer review of "A Review on Recent Trends and Applications of IoT in Additive Manufacturing"

_asi, doi:10.3390/asi6020050_

Round 1

Author Response

The paper describes the use of IoT in additive manufacturing and divides the application areas into five categories. The paper provides a very good overview of the state of the art and is well-written.

However, the concrete benefits of IoT need to be emphasized and discussed more strongly.

1) General

Point 1: The words "Review" should be added to the title of the article

Response 1: The authors have followed the reviewer’s advice and the word “Review” is included in the title of the article. A Review on Recent Trends and Applications of IoT in

Additive Manufacturing

Point 2: The abstract was too long. Please modify it to show the background, the challenge, the focus of this study, the novelty and the meaningful results.

Response 2: The abstract is modified as per the reviewer’s advice.

Point 3: The introduction must be revised. In the introduction, it is not clear what the problem or the research question is. Furthermore, it is not explicit how the rest of the paper is structured.

Response 3: Thank you. It was done.

Point 4: Additive manufacturing is also highly influenced by manual activities such as powder filling, component removal and process interruptions, so a machine operator is required. It is not clear how IoT can solve these sticking points or how these sticking points limit the potential of IoT.

Response 4: The authors have followed the reviewer’s advice and the suggestions are includedin the 4th paragraph of 2.5 section

Point 5: In Chapter 2, it is not clear in part what the concrete added value of IoT is supposed to be. The paper's altitude is too high.

Response 5: Thank you, Chapter has been revised and improved.

Point 6: Abbreviations such as NDE or GRA should be explained in the text.

Response 6: It was done.

Point 7: Chapter 3 can not only consist of a table but should also include a description. Thus, the table is not meaningful

Response 7: Thank you, it has been improved slightly.

2) Specific comments

Point 1:  The important sentence “One of the major and upcoming revolutions in the industry is integrating IoT with AM” was only supported by one source (line 69)

Response 1: Thank you. It was done.

Point 2: The source layout should be revised for multiple sources ([7], [8], [9] change to [7-9]) Line 82.

Response 2: Thank you. It was done.

Point 3: Lines 85 to 88: The terms of the individual additive manufacturing processes do not

correspond to the ASTM standard.

Response 3: Thank you for your insightful comment. It was corrected.

Point 4: Lines 229 through 231: no linkage between USAF and AM and DT.

Response 4: It was done.

Point 5: The added value of lines 319 to 325 is not clear

Response 5: It was corrected.

Point 6: 2.1: As a reader, it is not clear to me how the digital twin (DT) offers added value here. What is the difference to existing measures, e.g., control strategies? How is DT going to help with design? These are the exciting questions that need to be answered.

Response 6: Thank you for your valuable comments. It was added to the revised manuscript.

Point 7: Lines 337 to 339 should be described in more detail

Response 7: : It was corrected.

Point 8: What do bioprinters have to do with IoT in detail? Why should this manufacturing process be more suitable than others?

Response 8: Thank you. It was done.

Point 9: Line 458: Missing dot

Response 9: Thank you. It was done.

Point 10: Line 485: Capitalize the beginning of the sentence

Response 10: Thank you. It was done.

Point 11: Additively manufactured components can have high complexity and internal structures. Using 3D scanning, these geometries can only be partially identified and component defects inferred. Thus, a CT scan would be preferable rather than being able to use IoT profitably.

Response 11: The authors have followed the reviewer’s advice and

Point 12: Thank you. It was done.

Response 12: The authors have followed the reviewer’s advice and

Point 13: Thank you. It was done.

Response 13: The authors have followed the reviewer’s advice and

Reviewer 2 Report

The authors present a review that is aimed at the current issue of the use of modern tools for monitoring and remote control of electronic systems. They focus on the main trends and their application in additive manufacturing. They summarize the advantages and limitations of applications such as 3D bioprinting, 3D scanning, and human augmentation.

The presented study is beneficial, however, I list below some topics for editing and supplementing the text of the manuscript.

1) In my opinion, the text of section 2.5. INPUT PARAMETERS OPTIMIZATION needs to be corrected Some parts are not clear. I can't find an explanation of used abbreviations WAAM, NDE, GRA there. 

2) Next some links are not correct. For example line 552 [Manhunath], line 554 [Murali], line 404 [1g8].

Author Response

Comments and Suggestions for Authors

The authors present a review that is aimed at the current issue of the use of modern tools for monitoring and remote control of electronic systems. They focus on the main trends and their application in additive manufacturing. They summarize the advantages and limitations of applications such as 3D bioprinting, 3D scanning, and human augmentation.

The presented study is beneficial; however, I list below some topics for editing and supplementing the text of the manuscript.

Point 1: In my opinion, the text of section 2.5. INPUT PARAMETERS OPTIMIZATION needs to be corrected. Some parts are not clear. I cannot find an explanation of used abbreviations WAAM, NDE, GRA there.

Response 1: The authors have followed the reviewer’s advice and an explanation is now provided in the updated manuscript.

Point 2: Next some links are not correct. For example, line 552 [Manhunath], line 554 [Murali], and line 404 [1g8].

Response 2: Asper the reviewer’s advice the corrections are done and updated in the revised manuscript.

Reviewer 3 Report

This work is reviewing recent trends of IoT application in industry 4.0.

The work is good because it can be refence of the young researchers.

we recommend to cite very recent works published in 2023 , 2022,2021  and 2020... because the most refences are old.

Discuss other usage of IoT in industry...

Author Response

Comments and Suggestions for Authors

Point 1: This work is reviewing recent trends of IoT applications in industry 4.0.

Response 1: Authors would like to thank the reviewer for the positive feedback

Point 2: The work is good because it can be a reference for young researchers.

Response 2: Authors would like to thank the reviewer for the positive feedback

Point 3: we recommend citing very recent works published in 2023, 2022,2021 and 2020... because most references are old.

Response 3: Thank you, authors have followed the reviewer’s advice and recent articles were cited in the revised manuscript.

Point 4: Discuss other uses of IoT in the industry...

Response 4: Thank you for your insightful comment. Some of the other uses of IoT in the various industries were also included in the introduction part of the revised manuscript.

Round 2

Reviewer 2 Report

The authors accepted my previous comments and suggestions.
I accept the paper in present form.